# Entanglement parity effects in the Kane-Fisher problem

**Chunyu Tan[1], Yuxiao Hang[1,†], Stephan Haas[1] and Hubert Saleur[1,2]**

**1** Department of Physics and Astronomy,University of Southern California, Los Angeles, CA 90089-0484, USA

**2** Institut de Physique Théorique, Université Paris Saclay, CEA, CNRS, F-91191 Gif-sur-Yvette, France

† yhang@usc.edu

## Abstract

We study the entanglement of a segment of length $\ell$ in an XXZ chain with one free extremity and the other connected to the rest of the system with a weak bond. We find that the von-Neumann entropy exhibits terms of order $O(1)$ with strong parity effects, that probe the physics associated with the weakened bond and its behavior under the RG (Kane Fisher problem). In contrast with the XX case studied previously [1] the entropy difference $\delta S \equiv S^e - S^o$ gives rise now to a "resonance" curve which depends on the product $\ell T_B$, with $1/T_B$ a characteristic length scale akin to the Kondo length in Kondo problems [2]. The problem is studied both numerically using DMRG and analytically near the healed and split fixed points. Interestingly - and in contrast with what happens in other impurity problems [3] - $\delta S$ can, at least at lowest order, be tackled by conformal perturbation theory.

# 1   Introduction

The study of terms of $O(1)$ in entanglement in the presence of defects presents many challenges, and several questions - in particular, those concerning universality - remain unanswered to this day [4, 5]. On the other hand, terms of $O(1)$ in general have a potentially interesting physical meaning, for instance in the presence of topological phases or topological defects, and their study is of more than of purely academic interest.

The conformal field theory is widely used in calculating entanglement entropy [6], in an earlier work [1] we considered such terms of $O(1)$ for free fermions in the presence of a boundary and a conformal defect, and found parity effects that did not decay away from the boundary. A point made in [1] was that these effects hinted at the possibility of a topological phase in the SSH model. Similar parity effects - involving, however, extra zero modes - were studied recently in [7]. The purpose of the present paper is to extend the analysis of [1] to the interacting case, where the physics will turn out to be different, and involve in particular a renormalization group (RG) flow.

Specifically, the system we consider here is an XXZ spin chain of length $L$ with free boundary conditions at either end, and a modified coupling at position $\ell$ (see below for a fully accurate description of the model). In the absence of the boundary, this provides one of the many possible variants of the so-called Kane-Fisher problem. The latter is usually defined in terms of a Luttinger liquid coupled to an impurity, but of course the formulation in terms of spin chain is equivalent thanks to the Jordan-Wigner transformation and bosonization. In what follows, we will freely use one or the other of these languages. Without interactions - i.e., in the XX chain - the modified coupling induces, in the RG sense, an exactly marginal perturbation. The entanglement of the region $A$ starting at the boundary and ending at the modified bond was found in [1] to possess, on top of the expected $\ln \ell$ term with a factor proportional to the effective, coupling dependent central charge, a term of $O(1)$ that differs between the even and the odd cases. The corresponding difference $\delta S$ was found to be a universal function with interesting properties, interpolating between $\ln 2$ and $0$. In the case with interactions, the modified coupling induces an RG flow, with two possible fixed points, the fully split chain and the healed chain. Depending on the sign of $J_z$, one of these is stable and the other one unstable. Under a relevant perturbation of the stable fixed point by a modified coupling at position $\ell$, the entanglement now has a $\ln \ell$ term whose slope depends on an effective central charge, and can be expressed as a function of $\ell T_B$ where $T_B$ is a characteristic energy scale akin to the Kondo temperature in Kondo-like impurity problems. In this case again, we find below that the terms of $O(1)$ differ between the even and odd cases, and that the corresponding difference $\delta S$ is now a universal quantity depending, like the effective central charge, on the product $\ell T_B$. Just as in the non-interacting case $J_z = 0$, $\delta S$ interpolates between $\ln 2$ and $0$.

As commented in the conclusions, similar parity effects are observed in the periodic case. It is interesting to reflect a bit on their physical meaning. The best way to start is to think of

Kondo physics from the point of view of entanglement. It would be tempting to think that as the Kondo impurity gets screened by conduction electrons at low energy, it somehow becomes more entangled with them. But this clearly cannot be: the entanglement of the Kondo impurity as measured by the Von Neumann entropy of the corresponding spin with the rest of the system (the "single site impurity entanglement entropy" [8]) is (in the absence of a magnetic field) fixed at **ln 2** irrespective of the Kondo coupling. To be able to define non-trivial quantities (that can be used later on to provide signatures of the screening cloud [9]) one needs to introduce another (length) scale. In the Kondo literature, it is common to consider for this an interval of the system extending a distance $r$ from the impurity (in the s-wave language) [8,10], or, in related purely one-dimensional problems, an interval of length $L$ centered on the impurity [3]. The physics at play can then be understood in terms of valence bonds originating from the impurity and reaching out to the rest of the system. Roughly, the "impurity part" [8] of the entanglement reaches **ln 2** when $r \lesssim \xi_K$ (with $\xi_K$ the Kondo length) that is, when $r$ is smaller than the typical length of the valence bond originating from the impurity spin. The physics we have in our case is somewhat similar and could be intuitively interpreted in a valence bond picture [11]. Think for instance of the Hamiltonian (1) below. As illustrated in Fig. 1(a), in the limit of very small $\lambda$ (corresponding to a very small impurity bond) and in the simplified valence bond picture, valence bonds "prefer" not to stretch over the weak link: only one is forced to do so in the odd length case, contributing **ln 2** to the difference of entanglements between odd and even. On the contrary, if $\lambda \sim 1$, there are a lot of such valence bonds, and even though the parity of their numbers is odd in the odd case and even in the even case, the difference averages to a small term that decays with $L$. Fig. 1(b) shows the constant terms in even and odd cases when $J_z = 0$ which confirms our qualitative arguments (it is difficult to define unambiguously the $O(1)$ terms when $J_z \neq 0$, see below). In the intermediate region, the result depends on the average length of the valence bonds, which in this interpretation becomes $1/T_B$, the equivalent of $\xi_K$. Interestingly, while the underlying physics is the one of the Kane-Fisher [12,13] problem, from the point of view of entanglement we have results akin to the Kondo problem, with entanglement curves always interpolating between **ln 2** and **0**, and a "resonance" as can be seen e.g. in Fig. 4.

The paper is organized as follows. In section 2, we discuss the vicinity of the split fixed point and in section 3 the vicinity of the healed fixed point. In both cases, we provide "ab-initio" numerical results together with comparison with (conformal) perturbative calculations, especially of the entanglement. Most detailed calculations are done in subsections 3.3, 3.5 where some subtle aspects - including the renormalization between bare and renormalized couplings - are investigated in detail. We find in particular that, even though the entanglement cut and the location of the perturbation coincide, no non-universal effects seem to be encountered. This is in contrast with the current expectation for entanglement across topological defects [4].

## 2 Physics around the split fixed-point

### 2.1 Generalities

We consider first the Hamiltonian

$$
\begin{aligned}
H^A =\ & \sum_{j=1}^{\ell} \left( \sigma_j^x \sigma_{j+1}^x + \sigma_j^y \sigma_{j+1}^y + J_z \sigma_j^z \sigma_{j+1}^z \right) + \lambda \left( \sigma_\ell^x \sigma_{\ell+1}^x + \sigma_\ell^y \sigma_{\ell+1}^y + J_z \sigma_\ell^z \sigma_{\ell+1}^z \right) \\
& + \sum_{j=\ell+1}^{\infty} \left( \sigma_j^x \sigma_{j+1}^x + \sigma_j^y \sigma_{j+1}^y + J_z \sigma_j^z \sigma_{j+1}^z \right),
\end{aligned}
\tag{1}
$$

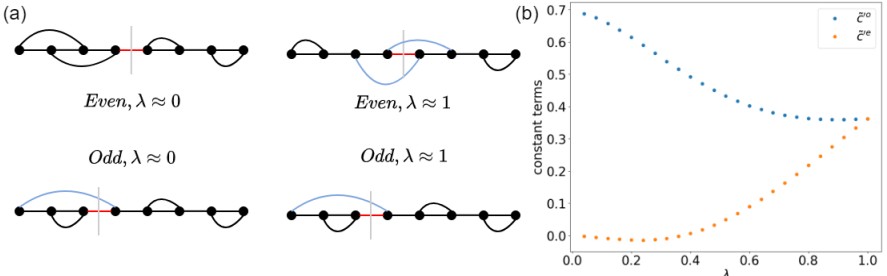

Figure 1: (a). Valence bond formation for $\lambda \approx 0$ ($\lambda \approx 1$) for both even and odd scenarios. The bonds separating the regions of the bipartition are illustrated by grey lines, and inter- (intra-) subsystems valence bonds are depicted by black (dark blue) curved lines. (b). The constant $O(1)$ terms for even and odd subsystems when $J_z = 0$

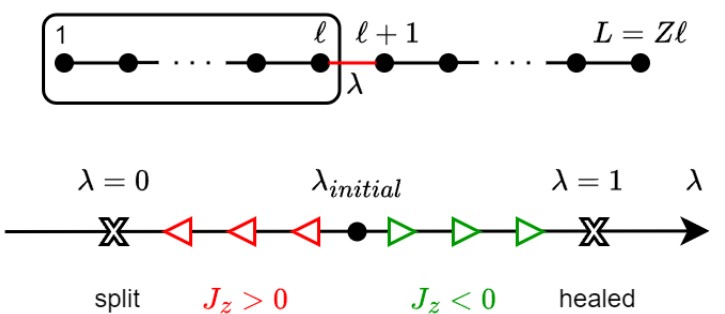

Figure 2: The different types of flows near the split fixed-point.

where the $\sigma$'s are Pauli matrices. Set

$$g = 2 - \frac{2}{\pi}\arccos(J_z),\tag{2}$$

so $g > 1$ for $J_z > 0$ and $g < 1$ for $J_z < 0$. Near the split fixed point $\lambda = 0$, a small nearest-neighbor interaction as in (1) is an operator of dimension $(length)^{-g}$. This means that

near the split fixed point $\lambda$ is relevant if $J_z < 0$

near the split fixed point $\lambda$ is irrelevant if $J_z > 0$. $\qquad(3)$

The RG flows are thus as in Fig. 2. Writing the perturbation as $\lambda\mathcal{O}$ this product must have dimension $(length)^{-1}$ and thus, if $\mathcal{O}$ has dimension $(length)^{-g}$, we find

$$\dim [\lambda] = (length)^{g-1}.\tag{4}$$

Hence we can construct a quantity of dimension $(length)^{-1}$ (a temperature) by considering[1]

$$T_B \equiv \lambda^{1/(1-g)}.\tag{5}$$

In the relevant case, the chain at large distances appears healed. This can be clearly seen if we consider the entanglement of the region of size $\ell$ to the left of the cut with the rest of the system: the leading ("bulk") behavior of the entanglement should interpolate from $S = 0$

---

[1]Note that definitions of $T_B$ differing by numerical factors may appear in the literature.

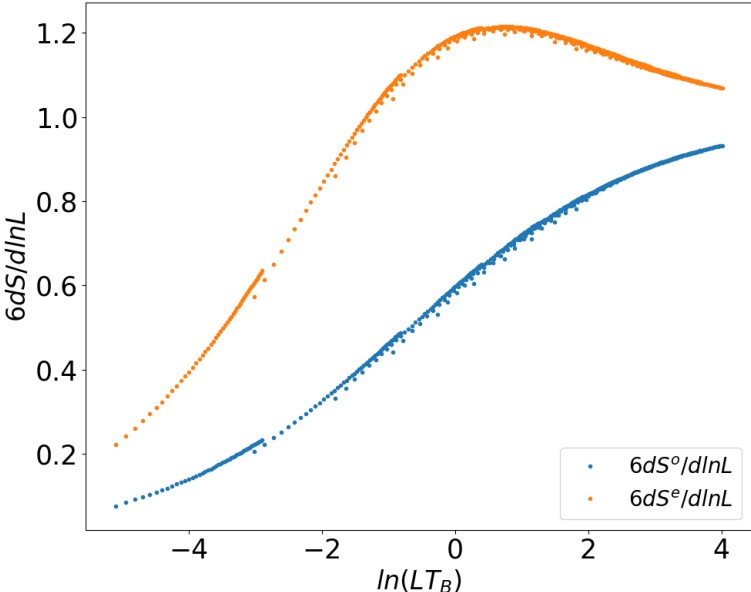

Figure 3: Crossover in the bulk entanglement entropy for Hamiltonian 1. $T_B$ is defined in (5).

to $S \approx \frac{1}{6}\ln\ell$ as $\ell$ is increased at fixed $\lambda$ (for earlier studies of this problem, see [14]. This is illustrated in Fig. 3 where we have plotted the derivative of the entanglement entropy for the system in the particular case $L = 2\ell$. Recall that, from finite size scaling results, the entanglement in the healed case always has the leading behavior $\frac{c}{6}\ln L$ with $c = 1$ here. This is seen in Fig. 3 as the the two curves go to $6 dS/d\ln L = 1$ in the IR. Note that the results look quite different for odd and even lengths $\ell$ (represented by $S^e$ and $S^o$ respectively). It is this difference we shall be interested in in what follows.

Like in [1] we expect to have, for an infinite system (more complicated formulas involving $\ell/L$ are required for a finite system, see below)

$$\frac{dS^e_{A,\text{imp+bdr}}}{d\ln\ell} = F(\ell T_B) + f^e(\ell T_B)$$
$$\frac{dS^o_{A,\text{imp+bdr}}}{d\ln\ell} = F(\ell T_B) + f^o(\ell T_B) \tag{6}$$

where $F, f^e, f^o$ are non-trivial, universal functions. Note that in [1], the forms (6) were encountered when considering a non-interacting system with a dot-like impurity - that is, two successive bonds modified. This case gave rise to a non-trivial RG flow, just like the case of a single modified bond in the presence of interactions that we study in this paper.

In (6), $F$ encodes a sort of effective central charge, while $f^{e,o}$ are "terms of $O(1)$". This name might sound inadequate from (6), but as discussed in detail in [15], only derivatives of the entanglement obey proper scaling, so the (parity independent) "bulk" term $\frac{c}{6}\ln\ell$ ceases to dominate the expression for the entanglement, once derivatives are taken, due to the fact that $\frac{d}{d\ln\ell}\ln\ell = 1$. Like in our earlier work [1] we shall focues on the difference $f^e - f^o$, which originates from terms in $S^{e,o}$ whose difference does not decay as $\ell \to \infty$.

In this context, $\lambda$ relevant means $\delta S \equiv S^e - S^o$ will evolve from something smaller than $\ln 2$ to $0$ as $L$ increases for a fixed $\lambda$ (that is, healing occurs) while irrelevant means $\delta S$ evolves from something smaller than $\ln 2$ to exactly $\ln 2$ as $L$ increases for a fixed $\lambda$.

## 2.2 The relevant case

We expect that in the limit of small $\lambda$ and large $L$, and *for the relevant case* ($g < 1$), results should have a universal dependency on the product $LT_B$. Note that this combination is small when $L$ or $\lambda$ is small, large when $L$ or $\lambda$ are large, and that, since results depend only on $LT_B$, increasing $L$ at fixed $\lambda$ in the scaling limit is like increasing $\lambda$ at fixed $L$: in other words, the long distance physics corresponds to healing. Scaling per se only occurs formally in the limits $\lambda \to 0, L \to \infty$ with the product $L\lambda^{1/(1-g)}$ finite.

While the phenomenology is well understood in general, we focus here on aspects of entanglement in the presence of a boundary that have not been studied before except in the special free fermion case $J_z = 0$ [1]. Results confirming the qualitative RG picture are given below. Our numerical results are obtained by using the density matrix renormalization group (DMRG) algorithm and the Tenpy package [16]. We plot the difference of entanglements with the subsystem starting at the boundary and ending in the middle of the modified bond (i.e. containing the spins $j = 1, \dots, j = \ell$) for the cases $\ell$ even and $\ell$ odd, and set $\delta S \equiv S^e - S^o$. The total system size is taken to be $L = Z\ell$, with $Z$ a factor taken to be $Z = 2$ unless otherwise specified.

We note that there are in fact two possible variants of the problem: (at least), since another Hamiltonian without a $J_z$ term at the impurity bond

$$
\begin{aligned}
H^B &= \sum_{j=1}^{\ell} \left( \sigma_j^x \sigma_{j+1}^x + \sigma_j^y \sigma_{j+1}^y + J_z \sigma_j^z \sigma_{j+1}^z \right) + \lambda \left( \sigma_\ell^x \sigma_{\ell+1}^x + \sigma_\ell^y \sigma_{\ell+1}^y \right) \\
&+ \sum_{j=\ell+1}^{\infty} \sigma_j^x \sigma_{j+1}^x + \sigma_j^y \sigma_{j+1}^y + J_z \sigma_j^z \sigma_{j+1}^z
\end{aligned}
\tag{7}
$$

could also be considered. Since the $J_z$ term is irrelevant, it should not affect the universal limit of our results, as we will see below.

We first give results for Hamiltonian (7) in Fig. 4. Totally identical results are obtained in the scaling limit for the Hamiltonian (1) as shown in Fig. 5. In particular, the value of $T_B$ is the same for the two curves. This is easily understood since the $\sigma_\ell^z \sigma_{\ell+1}^z$ term is irrelevant near the split fixed point [17]: while it affects corrections to scaling, it simply disappears in the limit $\lambda \to 0, \ell \to \infty$.

We note that in Fig. 5 the data corresponding to (7) is a little "fuzzy" while the two curves are slightly off for $LT_B \gtrsim 1$. This is due to the difficulty of reaching the scaling limit in the deep IR region, where values of $L$ unreachable by DMRG would, strictly speaking, be necessary. This is a familiar problem in the study of interacting systems. It practice, we take the small difference between the two curves in Fig. 5 as a measure of the uncertainty about the true location of the scaling curve.

Varying $Z$ does not change results much, even though of course the exact curve does, indeed, depend on $Z$ (see below for a detailed study near the fixed points). For the sake of brevity, we refrain from showing numerical results confirming this.

## 2.3 The irrelevant case

In this case we start from a small tunneling term but are driven at low-energy to the situation where the system is split. This can be seen in the fact that $LT_B$ increases at fixed $\lambda$ when increasing $L$ but increases at fixed $L$ when *decreasing* $\lambda$. Hence, large $L$ behaves like small $\lambda$, and the split-fixed point is reached at large distances. Going to small $LT_B$ is formally equivalent to increasing $\lambda$ and thus, one would hope, to getting closer to the healed fixed-point. However, in this limit, other irrelevant operators will start playing a role, and there is no chance to reach

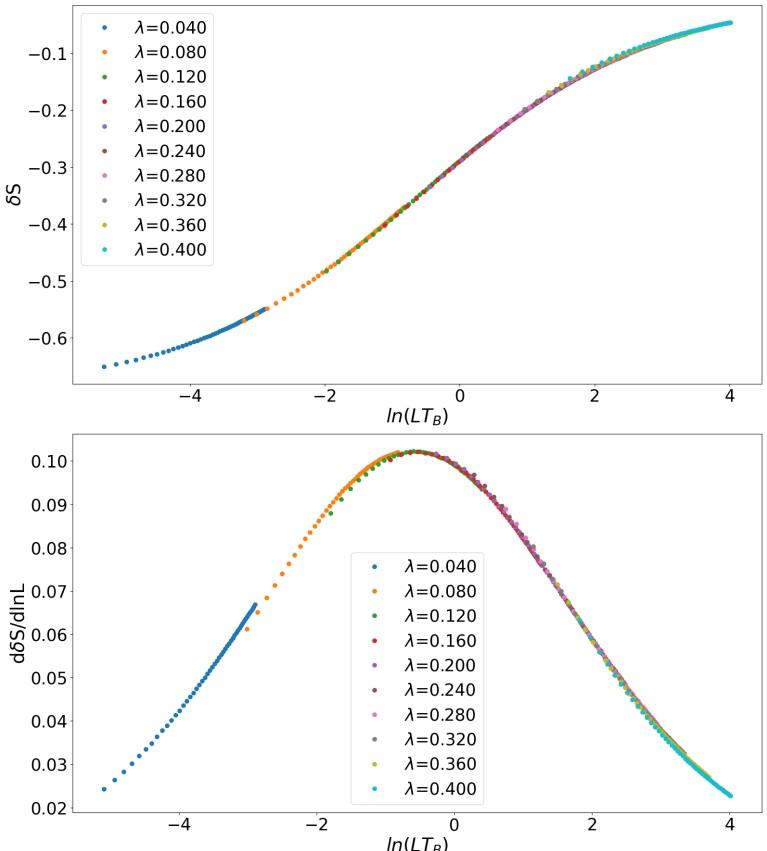

Figure 4: Healing flow for $J_z = -0.5$. Here, $Z = 2$, and the Hamiltonian is (1).

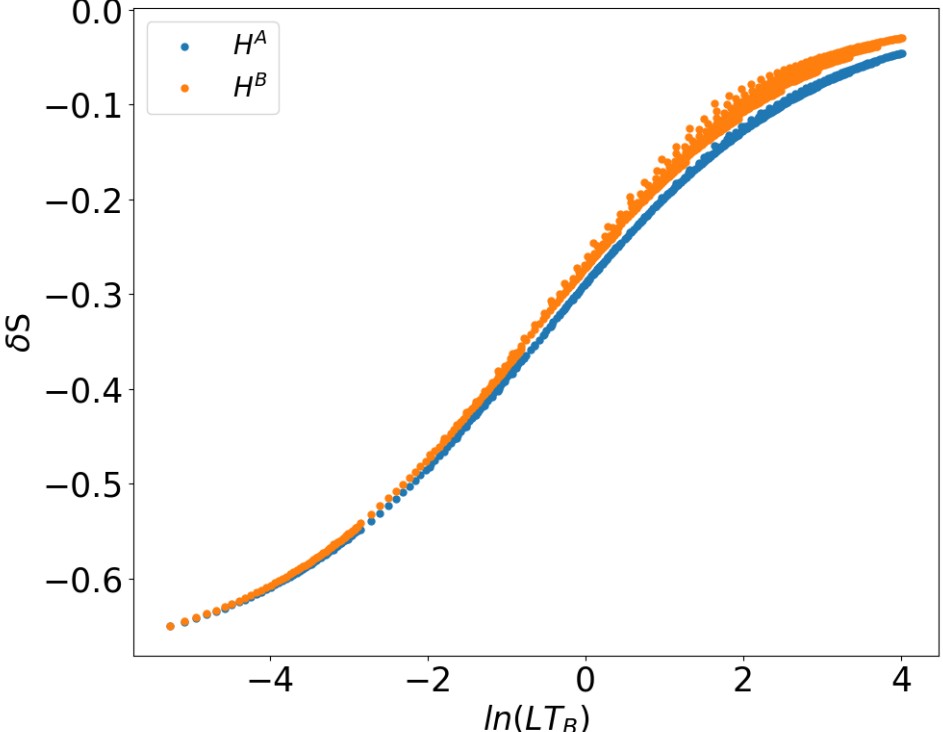

Figure 5: Healing flow for $J_z = -0.5$. Here, $Z = 2$, and the Hamiltonians (1) and (7) are compared. In the data shown, $\lambda$ is varied between values **0.04** and **0.4**

this fixed-point without fine tuning. In practice, this simply means that the left-hand side of the curves plotting $\delta S$ as a function of $LT_B$ are not fully universal. See Figs. 6 for some illustrations. In the following, we will mostly restrict to the study of relevant perturbations.

## 2.4 Some perturbative calculations

The first question to ask is how $\delta S$ varies as a function of $\lambda$ at small $\lambda$. In order to answer this question we need to think first about the situation at $\lambda = 0$, i.e. when the two systems are totally decoupled. The difference between even and odd is then spectacular. In the even case, both sides have an even number of spins and are in a (non-degenerate) ground-state of total spin $S^z = 0$ (we set $S^z = \frac{1}{2}\sum_j \sigma_j^z$). In contrast, in the odd case, both sides have a remaining spin $1/2$ degree of freedom, and thus have a ground-state degenerate twice, with $S_z = \pm 1/2$. This means in particular that the shift in energy due to the presence of the $\lambda \neq 0$ term exhibits different dependencies with $\lambda$ in the even and odd cases.

### 2.4.1 The shift in energy

For the even case, this shift can be obtained from non-degenerate perturbation theory and thus, by conservation of $S_z$, is quadratic in $\lambda$ at small coupling. In contrast, for the odd case, the shift comes from degenerate perturbation theory, and since states with spin $S_z = \pm 1/2$ have the same-energy, conservation of $S_z$ does not preclude the presence of a term linear in $\lambda$. It is interesting to push these considerations a bit further by using field theoretic techniques. First, we fermionize our spin chain (we will follow standard conventions such as those in [17]

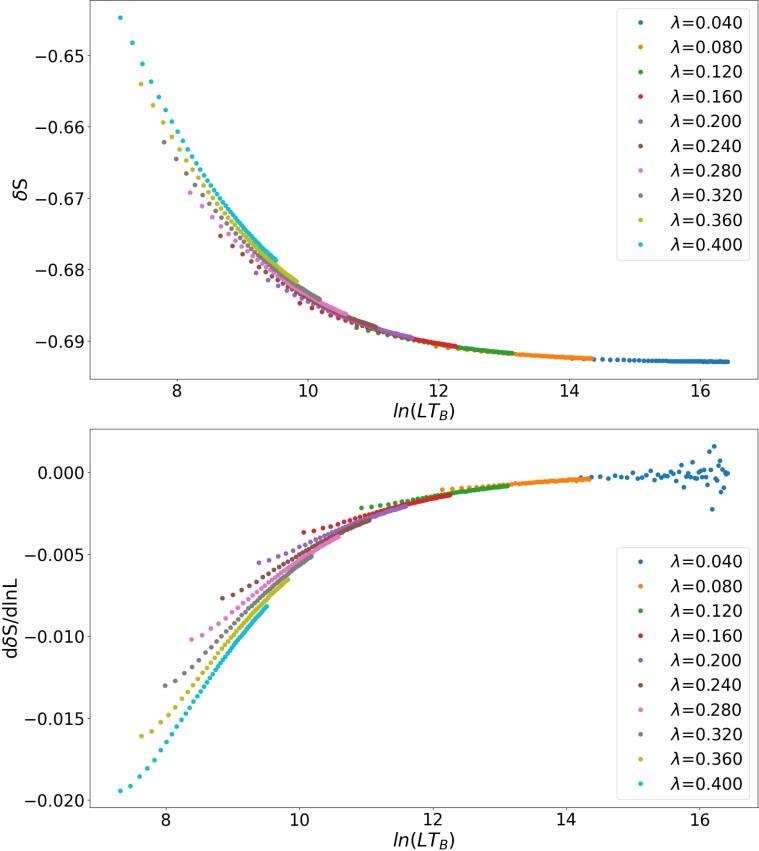

Figure 6: Flow for $J_z = 0.5$. Here, $Z = 2$, and the Hamiltonian is (7). The split fixed-point is recovered in the IR, while the healed fixed-point is not reached in the UV.

whenever possible), leading to the two possible Hamiltonians

$$H^A = \sum_{j=1}^{\ell} \left( c_j^\dagger c_{j+1} + \text{h.c.} + J_z (c_j^\dagger c_j - 1/2)(c_{j+1}^\dagger c_{j+1} - 1/2) \right) + \tag{8}$$

$$\lambda \left( c_\ell^\dagger c_{\ell+1} + \text{h.c.} + J_z (c_\ell^\dagger c_\ell - 1/2)(c_{\ell+1}^\dagger c_{\ell+1} - 1/2) \right)$$

$$+ \sum_{j=\ell+1}^{\infty} \left( c_j^\dagger c_{j+1} + \text{h.c.} + J_z (c_j^\dagger c_j - 1/2)(c_{j+1}^\dagger c_{j+1} - 1/2) \right) \tag{9}$$

and

$$H^B = \sum_{j=1}^{\ell} \left( c_j^\dagger c_{j+1} + \text{h.c.} + J_z (c_j^\dagger c_j - 1/2)(c_{j+1}^\dagger c_{j+1} - 1/2) \right) + \lambda \left( c_\ell^\dagger c_{\ell+1} + \text{h.c.} \right)$$

$$+ \sum_{j=\ell+1}^{\infty} \left( c_j^\dagger c_{j+1} + \text{h.c.} + J_z (c_j^\dagger c_j - 1/2)(c_{j+1}^\dagger c_{j+1} - 1/2) \right) \tag{10}$$

We recall the formulas for the decomposition of lattice fermions into continuous fields (see e.g. [17] for discussion of related problems)

$$c_j \mapsto e^{iK_F j} \psi_R + e^{-iK_F j} \psi_L \tag{11}$$

At half-filling, $K_F = \frac{\pi}{2}$. For a chain starting at $j = 1$, we have formally $c_j = 0$ for $j = 0$ to take into account the chain termination, so at that extremity the boundary conditions $\psi_L = -\psi_R$. Meanwhile, the conditions at the other extremity depend on the parity of the length. If the last site is $l$, we set

$$c_{l+1} \propto \psi_R + e^{-2iK_F(l+1)} \psi_L = 0 \tag{12}$$

so at half-filling this becomes

$$\psi_R(l+1) + (-1)^{l+1} \psi_L(l+1) = 0 \tag{13}$$

We see that if $l$ is odd we get the same boundary conditions at $l+1$ than at $0$, while if $l$ is even we get opposite boundary conditions.

Using bosonization formulas

$$\psi_R \propto \exp(i\sqrt{4\pi}\phi_R), \quad \psi_L \propto \exp(-i\sqrt{4\pi}\phi_L) \tag{14}$$

and handling the four-fermion term in the usual way [17] we get the continuum theory with bulk Hamiltonian

$$H = \frac{v}{2} \int dx \left[ \Pi^2 + (\partial_x \Phi)^2 \right] \tag{15}$$

The compactification radius of the boson is

$$R \equiv \sqrt{\frac{g}{4\pi}} \tag{16}$$

while the sound velocity is given by

$$v = \frac{\pi}{2} \frac{\sqrt{1 - J_z^2}}{\arccos J_z} \tag{17}$$

Note that $v = 1$ if $J_z = 0$. The boundary conditions at the origin in the non-interacting case are $\Phi = \phi_R + \phi_L = \frac{\pi}{\sqrt{4\pi}}$ which becomes $\Phi(0) = \pi R$ in general. From (13) it follows that, on

the right side we have $\Phi(l+1) = \pi R$ for $l$ odd and $\Phi(l+1) = 0$ for $l$ even (all these modulo $2\pi R$). So to summarize we can simply write

$$\Phi(0) = \pi R, \ \Phi(l+1) = 2\pi R S^z \tag{18}$$

with $S^z$ integer (half an odd-integer) for $l$ even (odd).

We now consider perturbation around the almost split fixed point, with Hamiltonian $H^A$ (1) or $H^B$ (7). To all orders in perturbation theory, the correlators we need to evaluate are factorized into correlators for two decoupled sub-systems - two open chains of equal length $\ell$ in our set-up with $Z = 2$. Let us now consider the shift in ground-state energy due to the tunneling. We start with the case $\ell$ even where the ground-state of either half is non degenerate. The Hamiltonian we must consider is

$$\begin{aligned}
H = &\frac{v}{2} \int_0^\ell dx \left[(\Pi^{(1)})^2 + (\partial_x \Phi^{(1)})^2\right] + \frac{v}{2} \int_\ell^{2\ell} dx \left[(\Pi^{(2)})^2 + (\partial_x \Phi^{(2)})^2\right] \\
&+ \lambda Z_\lambda \cos \frac{\beta}{\sqrt{2}} (\tilde{\Phi}^{(1)}(\ell) - \tilde{\Phi}^{(2)}(\ell))
\end{aligned} \tag{19}$$

where $Z_\lambda$ is the renormalization factor between the renormalized and the bare couplings (a thorough discussion of such factors is provided in the next section), and we have set

$$\beta = \sqrt{2\pi g} \tag{20}$$

Note that the tunneling term is expressed in terms of the dual field $\tilde{\Phi} = \phi_R - \phi_L$ since the field $\Phi$ takes a fixed value on either side of the tunneling bond.

To proceed, we need to introduce (imaginary) time, and thus to discuss propagators in the strip geometry. We need correlators of (exponentials of) the dual field $\tilde{\Phi}$ with Dirichlet boundary conditions (which are the same as correlators of (exponentials of) the field $\Phi$ with Neumann boundary conditions). Denoting by $y$ the imaginary time coordinate on a strip of width $\ell$, we find the propagator on the edge (i.e. for points on the right (or left) boundary) to be

$$\langle \tilde{\Phi}^{(i)}(y) \tilde{\Phi}^{(i)}(y') \rangle = -\frac{1}{2\pi} \ln \left| \frac{\ell}{\pi} \sinh \frac{\pi v}{\ell} (y - y') \right|^2 \tag{21}$$

so we have

$$\langle e^{i\frac{\beta}{\sqrt{2}}(\tilde{\Phi}^{(1)}(y) - \tilde{\Phi}^{(2)}(y))} e^{-i\frac{\beta}{\sqrt{2}}(\tilde{\Phi}^{(1)}(y') - \tilde{\Phi}^{(2)}(y'))} \rangle = \frac{1}{\left| \frac{\ell}{\pi} \sinh \frac{\pi v}{\ell} (y - y') \right|^{\frac{\beta^2}{\pi}}} \tag{22}$$

where one should note the apparition of $v$ on the right-hand side - due to the fact that the continuum limit of the lattice Hamiltonian is not isotropic in space/(imaginary)time.

Going back to the shift of the ground state-energy, the first order correction vanishes because, in the ground-state with Dirichlet boundary conditions, $\langle e^{i\beta\tilde{\Phi}} \rangle = 0$. To get the second order, we determine the partition function in an annulus geometry with the imaginary time length $\Lambda \gg \ell$. It follows that the shift in energy is proportional to the finite integral

$$\lambda^2 \frac{Z_\lambda^2}{2} \left(\frac{\ell}{\pi}\right)^{-2g} \int dy \frac{1}{|\sinh \frac{\pi v}{\ell} y|^{2g}} = \frac{\lambda^2 Z_\lambda^2}{v} \left(\frac{\ell}{2\pi}\right)^{1-2g} \frac{\Gamma(g)\Gamma(1-2g)}{\Gamma(1-g)}, \ g < \frac{1}{2} \tag{23}$$

so

$$\delta E = \lambda^2 c_1 \ell^{1-2g} \tag{24}$$

Since $\lambda \propto T_B^{1-g}$, this goes as $\ell^{-1}(\ell T_B)^{2-2g}$.

When $\frac{1}{2} < g < 1$, the integral is UV divergent. It can be regularized by the introduction of a small distance cut-off, adding to the right hand-side a term

$$\frac{\lambda^2}{v} \frac{Z_\lambda^2}{2} \left(\frac{\ell}{\pi}\right)^{-2g} \left(\frac{\ell}{\pi}\right)^{2g} \int_a \frac{dy}{y^{2g}} = \frac{\lambda^2}{v} \frac{Z_\lambda^2}{2} a^{1-2g}$$

and thus a non-universal contribution so we expect the change in energy

$$\delta E = \lambda^2 \left(c_1 \ell^{1-2g} + c_2\right) \tag{25}$$

where $c_1, c_2$ are constants. In the odd case the ground state is degenerate four times, since each half has a leftover spin $1/2$: we have thus $|\Omega\rangle_{\alpha\beta} = |\alpha\rangle \otimes |\beta\rangle$. In each of the subsystems, the raising/lowering spin at the extremity has non-zero matrix elements between $|\pm\alpha\rangle$. Carrying out degenerate perturbation theory we expect a shift in energy proportional to $\lambda$, whereas the non-zero matrix elements should scale as $L^{-g}$ by dimensional analysis. Hence in this case

$$\delta E = c_3 \lambda \ell^{-g} \tag{26}$$

These results have been checked numerically.

### 2.4.2   The entanglement

The perturbative computation of the entanglement near the split fixed point involves some technical aspects that are best discussed elsewhere. But we can still find out some important simple facts.

We start by considering the simplest case $Z = 2$ and $\ell$ even. When $\lambda = 0$, the two subsystems are in a ground-state which mimics the case $\ell = 2$: the spin on either side of the cut is up or down, and since the total $S^z$ for each subsystem vanishes, the remaining $\ell - 1$ spins have a total $S^z$ that is down or up. In other words, the ground state of each subsystem can we written

$$|0\rangle = \frac{|(+)-\rangle - |(-)+\rangle}{\sqrt{2}} \tag{27}$$

where $(+)$ and $(-)$ stand for the state of the remaining $\ell-1$ spins with this total magnetization. By $Z_2$ symmetry, $(-)$ is obtained from $(+)$ by flipping all spins.

Imagine now calculating the ground state in perturbation theory, and restrict for simplicity to the case of the Hamiltonian $H^B$. The perturbation $\lambda V \equiv \frac{\lambda}{2} \left(\sigma_\ell^+ \sigma_{\ell+1}^- + h.c.\right)$ acting only on the extremity spins of the two subsystems couples to eigenstates of the decoupled system where one side has spin one and the other spin minus one. Call the corresponding eigenstates $|(1)_n\rangle$ and $|(-1)_n\rangle$, with energy $E_n$. Similarly, two insertions of $V$ couple to eigenstates where both sides have vanishing spin. Call the corresponding eigenstates $|(0)\rangle_n$ (with $|(0)_0\rangle = |0\rangle$). To second order we have then

$$|\Omega\rangle = |0\rangle \otimes |0\rangle + \lambda \sum_{n,m} a_{nm} \left(|(1)_n\rangle \otimes |(-1)_m\rangle + |(-1)_n\rangle \otimes |(1)_m\rangle\right)$$
$$+ \lambda^2 \sum_n b_n \left(|(0)_n\rangle \otimes |0\rangle + |0\rangle \otimes |(0)_n\rangle\right) \tag{28}$$

where, e.g. from first order perturbation theory,

$$a_{nm} = \frac{1}{2} \frac{\langle (1)_n|(+)+\rangle \langle (-1)_m|(-)-\rangle}{(E_n - E_0)(E_m - E_0)} \tag{29}$$

Taking the trace on the second subspace of the full density operator $\rho = |\Omega\rangle\langle\Omega|$ gives the contribution to the reduced density operator

$$\rho_A \mathrm{Tr}_B \rho = (1 + \lambda^2(b_0 + b_0^*))|0\rangle\langle 0| + \lambda^2 \sum_{nm} A_{nm} (|(1)_n\rangle\langle(1)_m| + |(-1)_n\rangle\langle(-1)_m|)$$
$$+ \lambda^2 \sum_{n>0} b_n |(0)_n\rangle\langle 0| + b_n^*|0\rangle\langle(0)_n| \tag{30}$$

where $A_{nm} = \sum_p a_{np} a_{pm}^*$.

The reduced density matrix is thus the sum of three operators acting on different subspaces, and whose products are all vanishing. Symbolically we write

$$\rho_A = \mathrm{Tr}_B \rho = (1 + \lambda^2(b_0 + b_0)^*)|0\rangle\langle 0| + \lambda^2 A + \lambda^2 B \tag{31}$$

so that the ratio

$$R_p \equiv \frac{\mathrm{Tr}\,\rho_A^P}{(\mathrm{Tr}\,\rho_A)^p} \tag{32}$$

has the structure

$$R_p = \frac{(1 + \lambda^2(b_0 + b_0^*))^p + \lambda^{2p}\,(\mathrm{Tr}A^p + \mathrm{Tr}B^p)}{(1 + \lambda^2(b_0 + b_0)^* + \lambda^2\,(\mathrm{Tr}A + \mathrm{Tr}B))^p} \tag{33}$$

and we find the entanglement entropy

$$S = -\frac{d}{dp}R_P\Big|_{p=1} = -2|X|^2\lambda^2\left(-\frac{1}{2} + \ln|X|^2\lambda^2\right) \tag{34}$$

where $X$ is a coefficient simply following from the previous calculations. It follows from this discussion that the entanglement entropy in the even case has a leading term going as $\lambda^2 \ln \lambda$.

It is useful to summarize the foregoing discussion in more general terms. In the even case, the ground state of each of the two decoupled systems is non-degenerate and has spin $S_z = 0$. Since the full Hamiltonian commutes with the total spin

$$[H, S_A^z + S_B^z] = 0 \tag{35}$$

the reduced density matrix $\rho_A$ commutes with the spin $S_A^z$ [18]:

$$[\rho_A, S_A^z] = 0 \tag{36}$$

Right at the decoupled point, $\rho_A$ only has matrix elements between the factorized ground-state and itself, both at $S_A^z = 0$. However, under the tunneling perturbation, the new ground-state acquires components onto states which, while having total spin $S_A^z + S_B^z = 0$, have $S_A^z = \pm 1$. After tracing over the $B$ degrees of freedom, and writing $\rho_A$ in block diagonal form with blocks labelled by $S_A^z$, this means that, two second order in perturbation theory, we have the structure

$$\rho_A = \mathrm{Tr}_B \rho = \begin{pmatrix} \boxed{\rho^{(0)}} & 0 & 0 & \cdots \\ 0 & \boxed{\rho^{(1)}} & 0 & \cdots \\ 0 & 0 & \boxed{\rho^{(-1)}} & \cdots \\ \cdots & \cdots & \cdots & \cdots \end{pmatrix} \tag{37}$$

where the labels refer to values of $S_A^z$ and the dots on the diagonal contain blocks of higher charge. The crucial point now is that $\rho^{(\pm n)}$ for $n \neq 0$ is a contribution of order $\lambda^{2n}$ since it takes $n$ actions of the perturbation to produce a state with spin $S_z = \pm n$ starting from a

state of vanishing spin. We thus see immediately that we can expect a structure as in (37) and consequently, after calculating the Reny entropy and taking the limit $p \to 1$, generate the leading term $\lambda^2 \ln \lambda$. Note meanwhile that, were we to carry out the perturbative expansion to higher orders, only terms *even* in $\lambda$ would be encountered.

The odd case is a bit different. Exactly at $\lambda = 0$ there is a potential ambiguity since the left and right hand sides are now both degenerate twice. The entanglement is not even defined at this point without specification of the state of the full system. However, as soon as $\lambda \neq 0$, this degeneracy is broken, and the ground state becomes unique and has $S_z = 0$. It is easy to identify this state in the case $Z = 2$, where the system with or without perturbation is symmetric under exchanges of the two sides and conserves the total spin: the ground state at finite $\lambda$ remains in the sector antisymmetric under the exchange, and with $S_z = 0$.

When $\lambda = 0$ we can write the (normalized) ground states of each side as combinations

$$
\begin{aligned}
|+\rangle &= \lambda_+^+ |(0)+\rangle + \lambda_-^+ |(++)-\rangle \\
|-\rangle &= \lambda_-^- |(0)+\rangle + \lambda_+^- |(--)+\rangle
\end{aligned}
\tag{38}
$$

where once again $(0), (++), (--)$ stand for states of the remaining $\ell - 1$ spins. We then choose the ground state of the whole system to be

$$
|\Omega\rangle = \frac{|+-\rangle - |-+\rangle}{\sqrt{2}}
\tag{39}
$$

The density matrix of the lhs then reads schematically, in the basis (38)

$$
\mathrm{Tr}_B \rho = \frac{1}{2}
\begin{pmatrix}
(\lambda_+^+)^2 & \lambda_+^+ \lambda_-^+ & 0 & 0 \\
\lambda_+^+ \lambda_-^+ & (\lambda_-^+)^2 & 0 & 0 \\
0 & 0 & (\lambda_+^-)^2 & \lambda_+^- \lambda_-^- \\
0 & 0 & \lambda_+^- \lambda_-^- & (\lambda_-^-)^2
\end{pmatrix}
\tag{40}
$$

Using that the ground states (38) are normalized we can easily calculate from this the Reni entropy and show that it gives, as expected, rise to $S = \ln 2$.

Now going through the same charge conservation arguments as before, we see that, when perturbing the ground state we will have to carry out a calculation similar to the one of the even case, resulting in a charge resolved structure for the density matrix now of the type

$$
\mathrm{Tr}_B \rho =
\begin{pmatrix}
\boxed{\rho^{(1/2)}} & 0 & 0 & 0 & \dots \\
0 & \boxed{\rho^{(-1/2)}} & 0 & 0 & \dots \\
0 & 0 & \boxed{\rho^{(3/2)}} & 0 & \dots \\
0 & 0 & 0 & \boxed{\rho^{(-3/2)}} & \dots \\
\dots & \dots & \dots & \dots & \dots
\end{pmatrix}
\tag{41}
$$

where the terms $\rho^{(\frac{1}{2}+n)}$ will come with factors $\lambda^{2n}$. Once again we will get in the end terms that are even in $\lambda$, with a leading correction going as $\lambda^2 \ln \lambda^2$.

Numerical results fully confirm this picture (see figure 7). We also give below a determination of the slopes of the leading terms, although we do not have a full analytical derivation at this stage. To conclude, we see that, in contrast with the energy, the entanglement at small $\lambda$ behaves similarly (but not identically) in the even and odd cases.

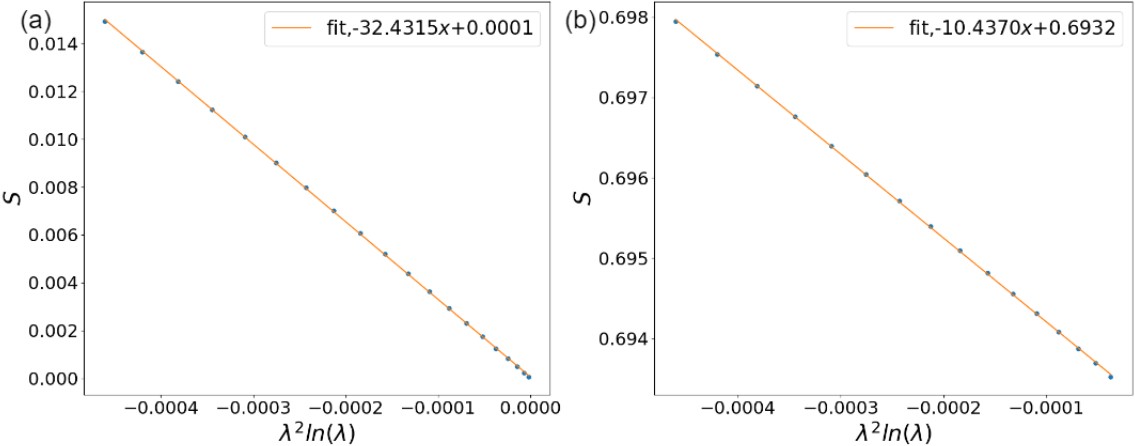

Figure 7: Study of the entanglement at small $\lambda$, $J_z$=-0.5, L=400 for Hamiltonian (1), confirming the dependency $S \propto -\lambda^2 \ln|\lambda|$. (a) Even case (b) Odd case.

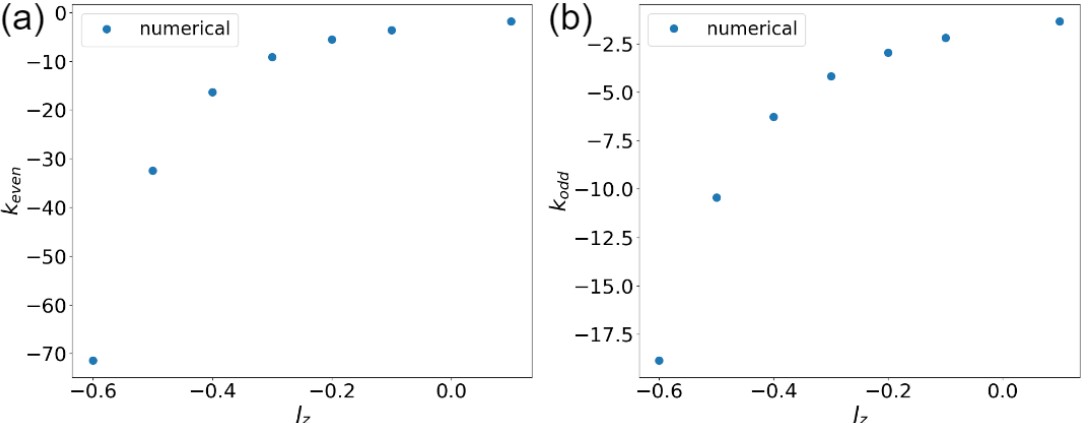

Figure 8: The slopes of the $\lambda^2 \ln|\lambda|$ leading term as a function of $J_z$ for Hamiltonian (1). (a) Even case (b) Odd case.

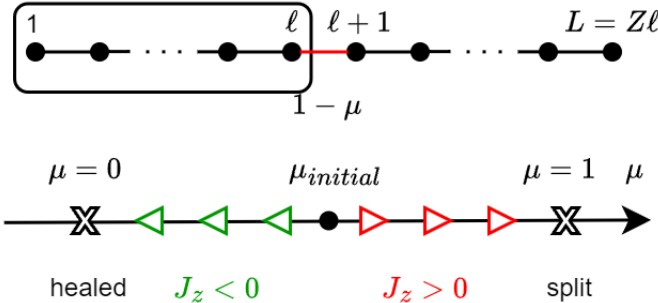

Figure 9: The different types of flows near the homogeneous fixed-point.

## 3 Physics around the homogeneous fixed-point

### 3.1 Generalities

We can also consider the vicinity of the homogeneous (uniform) fixed point. In this case the relevant Hamiltonians are

$$
\begin{aligned}
H^A &= \sum_{j=0}^{\ell}\left(\sigma_j^x\sigma_{j+1}^x + \sigma_j^y\sigma_{j+1}^y + J_z\sigma_j^z\sigma_{j+1}^z\right) + (1-\mu)\left(\sigma_\ell^x\sigma_{\ell+1}^x + \sigma_\ell^y\sigma_{\ell+1}^y + J_z\sigma_\ell^z\sigma_{\ell+1}^z\right) \\
&+ \sum_{j=\ell+1}^{\infty}\sigma_j^x\sigma_{j+1}^x + \sigma_j^y\sigma_{j+1}^y + J_z\sigma_j^z\sigma_{j+1}^z
\end{aligned}
\tag{42}
$$

and

$$
\begin{aligned}
H^B &= \sum_{j=0}^{\ell}\left(\sigma_j^x\sigma_{j+1}^x + \sigma_j^y\sigma_{j+1}^y + J_z\sigma_j^z\sigma_{j+1}^z\right) + (1-\mu)\left(\sigma_\ell^x\sigma_{\ell+1}^x + \sigma_\ell^y\sigma_{\ell+1}^y\right) + J_z\sigma_\ell^z\sigma_{\ell+1}^z \\
&+ \sum_{j=\ell+1}^{\infty}\sigma_j^x\sigma_{j+1}^x + \sigma_j^y\sigma_{j+1}^y + J_z\sigma_j^z\sigma_{j+1}^z
\end{aligned}
\tag{43}
$$

Perturbing the coupling near the uniform fixed point corresponds in the continuum limit to an operator of dimension $(\text{length})^{-g^{-1}}$ (together with an operator of dimension $(\text{length})^{-2}$, see below). We see that the regions of relevance and irrelevance are switched with respect to the previous section, and

near the homogeneous fixed point $\mu$ is irrelevant if $J_z < 0$

near the homogeneous fixed point $\mu$ is relevant if $J_z > 0$      (44)

The corresponding flows are sketched in Fig. 9. We see now that

$$
\dim \mu = (\text{length})^{g^{-1}-1}
\tag{45}
$$

Using the same kind of scaling argument as for the case of an almost split chain, we now expect the properties to have a universal dependency on $L\Theta_B$ with

$$
\Theta_B \equiv \mu^{1/(1-g^{-1})}
\tag{46}
$$

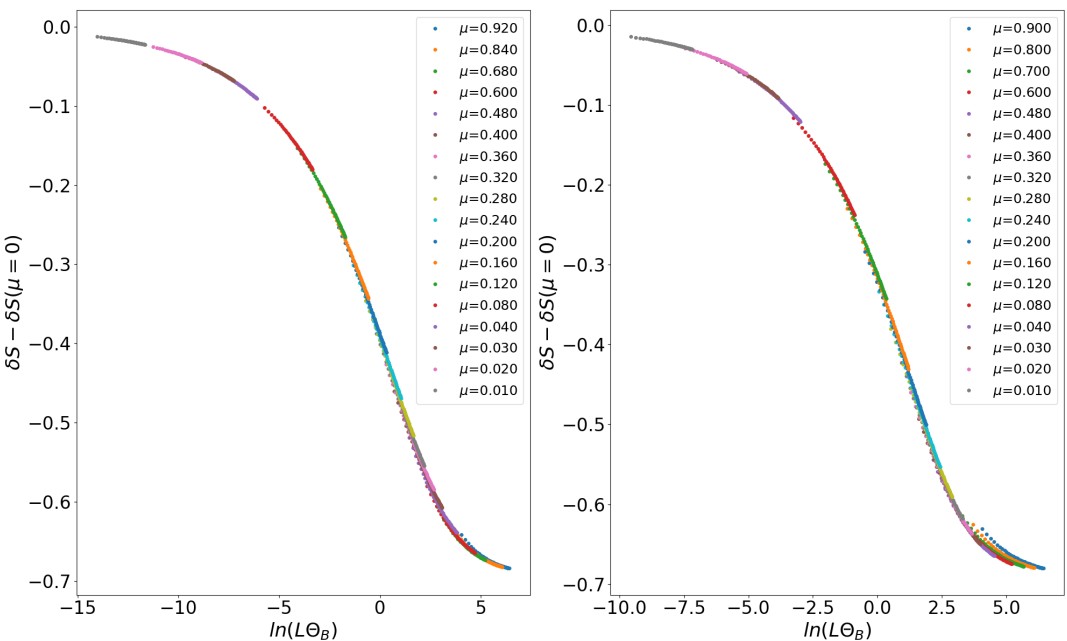

Figure 10: Flow for $J_z = 0.5, 0.7$. Here, $Z = 2$, and the Hamiltonian is (42). The weakly perturbed chain flows to the split fixed-point in the IR.

## 3.2   Numerics for the entanglement

In the relevant case, we now flow from the homogeneous to the split fixed point - this is illustrated in Fig. 10

## 3.3   Some perturbative calculations

We once again turn to bosonization. This time, since we start with a homogeneous chain, we have to consider a single bosonic theory on the half-line (or a segment of length $Z\ell$ if the system is finite). We start with (43) and need the crucial bosonization formula

$$\sigma_j^x \sigma_{j+1}^x + \sigma_j^y \sigma_{j+1}^y \;=\; 2(-1)^j c_1^\pm \cos \frac{\Phi}{R}(x = j) \tag{47}$$

where $c_1^\pm$ is a constant equal to $\frac{1}{\pi}$ in the non-interacting case, but otherwise not known exactly (see below). Note we only represented the leading term, of dimension $\frac{1}{4\pi R^2} = g^{-1}$. The next term would be proportional to $\left(\frac{\partial \Phi}{\partial x}\right)^2$, of dimension 2. It thus becomes the most relevant one when $g^{-1} = 2$, that is $J_z < -\frac{\sqrt{2}}{2}$. We will not study this region for now. In the non-interacting case $g = 1$, $R = \sqrt{4\pi}$ and we recover the results used in [1].

The Hamiltonian corresponding to (43) is then

$$H = \frac{v}{2} \int_0^\infty dx \left[ \Pi^2 + (\partial_x \Phi)^2 \right] - \frac{\mu_R}{\pi}(-1)^\ell \cos \frac{\Phi}{R}(x = \ell) \tag{48}$$

Note that in (48) we have introduced a *renormalized coupling constant* $\mu_R$. Indeed, while in the non-interacting case $\mu_R = \mu$ as defined with the lattice Hamiltonian (43) (which is the same as (42) in this case), in the presence of interactions, renormalization effects lead to $\mu_R = Z_\mu \mu$. The constant $Z_\mu$ - essentially the proportionality constant $c_1^\pm$ in (47) - is not

universal. Its value for the XXZ spin chain is not known exactly, but has been determined numerically to a great accuracy in [19] (for earlier work see [20]). We will use the values deduced from Table I in [19], after the correspondence $Z_\mu^B = \sqrt{8\pi^2 B_1^\pm} = \sqrt{4\pi^2} c_1^\pm$, while $J_z = \Delta$ (so that $Z_\mu^B = 1$ for $J_z = 0$). In fact, it will turn out that the relevant quantity is the ratio $\frac{Z_\mu^B}{\nu}$, where $\nu$ is given in (17)

For the Hamiltonian (42), we have to take into account the fact that $\sigma_j^z \sigma_{j+1}^z = c_1^z(-1)^j \cos \frac{\Phi(x=j)}{R}$, to leading order as well, for $J_z > -\frac{\sqrt{2}}{2}$. In this region, the Hamiltonian in the continuum limit if still (48) but now with a different value of the renormalization constant $Z_\mu^A$, since

$$\sigma_j^x \sigma_{j+1}^x + \sigma_j^y \sigma_{j+1}^y + J_z \sigma_j^z \sigma_{j+1}^z = (-1)^j (2c_1^\pm + c_1^z) \cos \frac{\Phi}{R}(x=j) \tag{49}$$

It follows that the new renormalization constant is $Z_\mu^A = \sqrt{8\pi^2}\left(\sqrt{B_1^\pm} + \frac{J_z}{2}\sqrt{B_1^z}\right)$. After dividing by $\nu$, this gives rise to the values listed below.

We now use these results to study the entanglement. The strategy is the same as the one used in [1]. What is called $(\lambda - 1)$ in eq. 43 ( in this section, equation numbers from reference [1] will be indicated by eq. ) is called $\mu$ here, while in eq. 44 we have $\beta = \frac{1}{R}$, so $\frac{\beta^2}{4\pi} = g^{-1}$ and $h = \frac{1}{2}g^{-1}$. Everything up until eq. 52 works as well in our case, although now we have

$$\left\langle \cos \frac{\Phi}{R}(w, \bar{w}) \right\rangle_{R_p} = \left(\frac{2\ell}{p}\right)^{2h} \frac{[\nu^2\tau^2[\nu^2\tau^2 + 4\ell^2]^{h\left(\frac{1}{p}-1\right)}}{[(\nu^2\tau^2 + 4\ell^2)^{1/p} - (\nu\tau)^{2/p}]^{2h}} \tag{50}$$

The asymptotic behaviors at large distance are now $\left\langle \cos \frac{\Phi}{R}(w, \bar{w}) \right\rangle_{R_p} \approx (2\ell)^{-2h}$, $\tau >> \ell$ and $\left\langle \cos \frac{\Phi}{R}(w, \bar{w}) \right\rangle_{R_p} \approx \left[\frac{1}{p}(2\ell)^{-1/p}\tau^{\frac{1}{p}-1}\right]^{2h}$, $\tau << \ell$. Like for when $h = \frac{1}{2}$, the resulting integral is still convergent - in fact, thanks to the subtraction coming from the denominator, it turns out to be *always convergent*! Setting $\nu\tau = 2\ell \tan\theta$ we have, replacing eq. 54:[2]

$$\frac{R_p(\mu_R)}{R_p(\mu_R = 0)} = 1 + \frac{2\mu_R}{\pi\nu}(2\ell)^{1-2h} \int_0^{\pi/2} \frac{d\theta}{\cos^2\theta}\left[p^{1-2h}\frac{(\sin\theta)^{2h(\frac{1}{p}-1)}\cos^{4h}\theta}{[1-(\sin\theta)^{2/p}]^{2h}} - p\right]$$

$$\equiv 1 + \frac{2\mu_R}{\pi}(2\ell)^{1-2h} I_p \tag{51}$$

Like in [1] we get the correction to the entanglement by

$$S = \frac{1}{6}\ln\left(\frac{\ell}{a}\right) + \frac{2\mu_R}{\pi\nu}(2\ell)^{1-2h} \left.\frac{d}{dp}I_p\right|_{p=1} \tag{52}$$

Remarkably, the resulting integral differs from the one when $h = \frac{1}{2}$ by a simple factor: $\left.\frac{d}{dp}I_p\right|_{p=1}(h) = 2h \left.\frac{d}{dp}I_p\right|_{p=1}(h = \frac{1}{2})$, [3] and we find in the end

$$S = \frac{1}{6}\ln\left(\frac{\ell}{a}\right) + \frac{1}{3}g^{-1}\frac{\mu_R}{\nu}(2\ell)^{1-g^{-1}} \tag{53}$$

---

[2]There is an unfortunate typo in eq. 54 of [1]: two of the cosines in the bracket, should be sinuses, as can be seen by setting $h = \frac{1}{2}$ in (51). Also, notice that $\mu$ in [1] is equal to $\frac{\mu}{\pi}$ in the present paper.

[3]It converges at $\theta = \frac{\pi}{2}$ in all cases.

where we used that $2h = g^{-1}$ and $\frac{d}{dp}I_p\big|_{p=1}(h = \frac{1}{2}) = \frac{\pi}{6}$. As in [1] the result only holds for $L = \infty$. When the ratio $\frac{\ell}{L}$ is finite, finite size effects have to be taken into account.

Comparing odd and even cases amounts to changing the sign of $\mu$ as discussed in [1]. This leads immediately to

$$\delta S = \frac{2}{3}g^{-1}\frac{\mu_R}{\nu}(2\ell)^{1-g^{-1}} \tag{54}$$

In the non-interacting case $J_z = 0$ we have $g = 1$ and we find $\delta S = \frac{2}{3}\mu$ like in eq. 58 of that reference. When the system has finite size $L = 2\ell$ (so $Z = 2$), we find, generalizing eq. 60[4]

$$\begin{aligned}
\delta S_{Z=2} &= .636779\, g^{-1}\frac{\mu_R}{\nu}\left(\frac{4\ell}{\pi}\right)^{1-g^{-1}} \\
&= .636779\, g^{-1}\frac{Z_\mu}{\nu}\mu\left(\frac{4\ell}{\pi}\right)^{1-g^{-1}}
\end{aligned} \tag{55}$$

As mentioned above we now observe that the integrals encountered in this calculation are *always convergent*, irrespective of the relevance of the perturbation. It follows that (55) should hold as well when the perturbation is relevant, i.e. when $J_z > 0$. The numerics indeed do not see anything happening when $J_z = 0$ is crossed.

On the other hand, result (55) only makes sense when the hopping term is the leading (ir)relevant operator. As $J_z$ crosses the value $-\frac{\sqrt{2}}{2}$, the term of ($J_z$ independent) dimension 2 dominates, and thus (55) ceases to be valid.

## 3.4 Comparison with numerics

The numerical analysis is a little tricky because, even in the absence of a local perturbation, the entanglement is known to already exhibit an alternating dependency upon $\ell$ [21–23], leading to

$$\delta S_{Z=2}(\mu = 0) = a(g)l^{-g^{-1}} \tag{56}$$

This correction is well identified in the literature, and the exponent usually written as $K$, the Luttinger constant, with $K = \frac{\pi}{2(\pi - \arccos J_z)} = \frac{1}{g}$. It is due to the leading irrelevant *bulk* oscillating term in the chain. We have first checked the result (56), as illustrated in Fig. 11 (a)

To leading order, we expect the correction (56) and the correction induced by the $\mu \neq 0$ perturbation to simply add up, so we should have

$$\delta S_{Z=2}(\mu) - \delta S_{Z=2}(\mu = 0) = .636779\, g^{-1}\frac{Z_\mu}{\nu}\mu\left(\frac{4\ell}{\pi}\right)^{1-g^{-1}} \tag{57}$$

We have therefore studied in what follows the quantity $\delta S_{Z=2}(\mu) - \delta S_{Z=2}(\mu = 0)$. Measures of the exponent obtained by plotting $\ln[\delta S_{Z=2}(\mu) - \delta S_{Z=2}(\mu = 0)] - \ln\ell$ for small values of $\mu$ give excellent, $\mu$ independent results, as illustrated in Fig. 11(b): To obtain results for the slope itself, we fit $\delta S_{Z=2}(\mu) - \delta S_{Z=2}(\mu = 0)$ for a series of values of $J_z$ - it turns out the relevant region involves values of $\mu$ as small as $5\,10^{-4}$. An example of such fit is given in Fig. 12(a).

The resulting slopes are then compared with the best known numerical values in Figs. 12(b) and 12(c) for the two possible Hamiltonians. Note the excellent agreement both in the relevant and irrelevant case as long as $J_z$ is not too close to $\pm 1$.

---

[4]The substitution for the $\ell$ dependent factor is $2\ell \to \frac{2L}{\pi}\sin\frac{\pi\ell}{L}$, so $2\ell \to \frac{4\ell}{\pi}$ when $L = 2\ell$. Otherwise, finite size gives rise to the same modified integral as in [1].

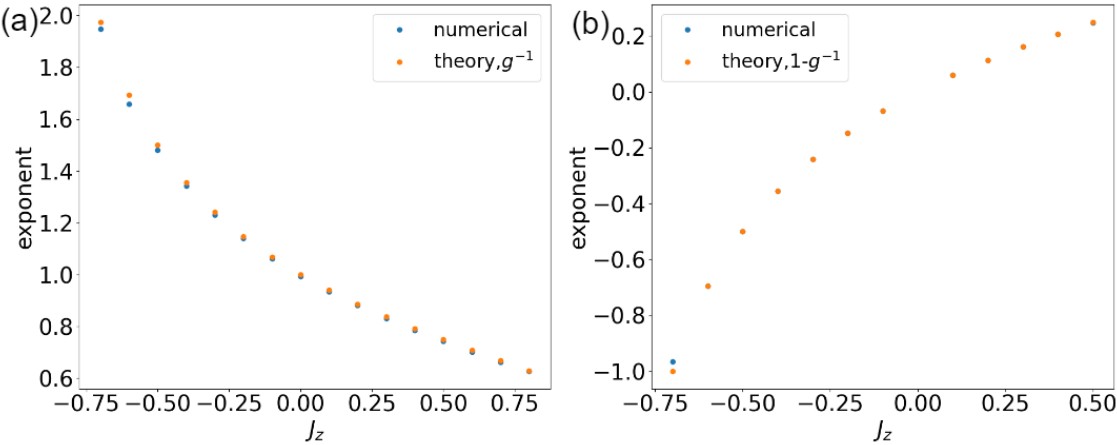

Figure 11: (a) Study of the exponent in(56). (b) Study of the exponent in (57).

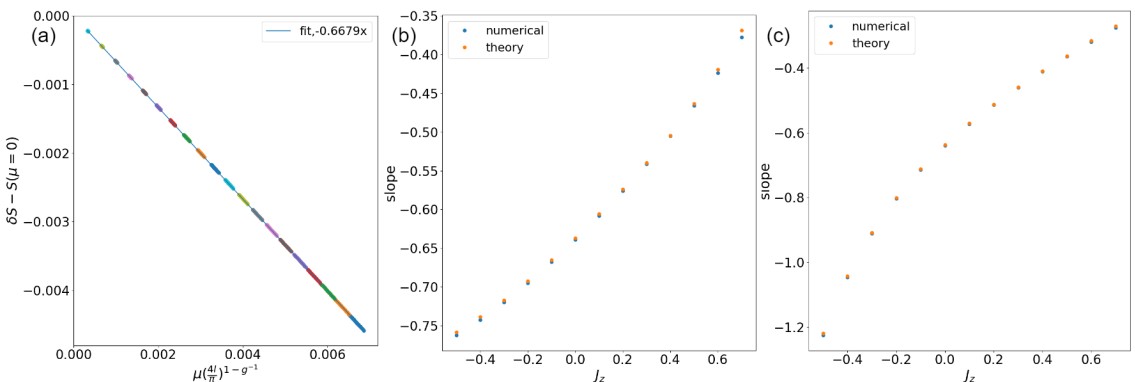

Figure 12: (a) Example of fit. (b). Study of the slope of $\delta S$ near the homogeneous fixed point for Hamiltonian (42).(c) Study of the slope of $\delta S$ near the homogeneous fixed point for Hamiltonian (43)

### 3.5   Determination of the renormalization factors

Instead of relying on the literature, we can of course determine $Z_\mu$ directly by studying the energy of the model. Indeed, using Hamiltonian (48) together with the result

$$\langle \cos \frac{\Phi}{R} \rangle = \frac{1}{(2\ell)^{2h}} \tag{58}$$

(recall $2h = g^{-1}$) leads immediately to the term $O(1)$ in energy

$$E^{(1)} = -\frac{\mu_R}{\pi} \frac{(-1)^\ell}{(2\ell)^{2h}} \tag{59}$$

and thus to the difference between even and odd

$$\delta E = -2\frac{\mu_R}{\pi} \frac{1}{(2\ell)^{2h}} \tag{60}$$

For a finite system we obtain the corresponding shift using a conformal mapping. If the total system is of length $L$ we have then

$$\langle \cos \frac{\Phi}{R}(\ell) \rangle = \left(\frac{\pi}{L}\right)^{2h} \frac{1}{[2\sin \frac{\pi\ell}{L}]^{2h}} \tag{61}$$

| $J_z$ | $Z_\mu^A/v$ | our num. $Z_\mu^A/v$ | $Z_\mu^B/v$ | our num. $Z_\mu^B/v$ |
|---|---|---|---|---|
| 0.700 | 0.865 | 0.886 | 0.636 | 0.646 |
| 0.600 | 0.928 | 0.938 | 0.700 | 0.708 |
| 0.500 | 0.970 | 0.976 | 0.760 | 0.763 |
| 0.400 | 1.000 | 1.001 | 0.811 | 0.815 |
| 0.300 | 1.012 | 1.015 | 0.861 | 0.864 |
| 0.200 | 1.017 | 1.020 | 0.908 | 0.911 |
| 0.100 | 1.012 | 1.016 | 0.954 | 0.957 |
| 0.000 | 1.000 | 1.003 | 1.000 | 1.004 |
| -0.100 | 0.978 | 0.982 | 1.047 | 1.051 |
| -0.200 | 0.948 | 0.952 | 1.097 | 1.100 |
| -0.300 | 0.908 | 0.911 | 1.150 | 1.153 |
| -0.400 | 0.856 | 0.861 | 1.208 | 1.212 |
| -0.500 | 0.794 | 0.798 | 1.277 | 1.282 |

Table 1: The table compares our results with the best numerical estimates [19]$Z_\mu^{A,B}/v$ for Hamiltonian (42) and Hamiltonian (43)

and thus for our case $Z = 2$ we find finally

$$\delta E = -2\frac{\mu_R}{\pi}\left(\frac{\pi}{4\ell}\right)^{2h} \tag{62}$$

From the definition $\mu_R = Z_\mu \mu$ a numerical determination of $\delta E$ gives access to the renormalization factor (recall $Z_\mu = 1$ for $J_z = 0$).

Note that this time the sound velocity $v$ does not enter. The values of $Z_\mu$ for Hamiltonians (42) (43) determined this way are given below in table 2, and compared with those from [19], with excellent agreement.

| $J_z$ | $Z_\mu^A$ | numerical $Z_\mu^A$ | $Z_\mu^B$ | numerical $Z_\mu^B$ |
|---|---|---|---|---|
| 0.700 | 1.220 | 1.222 | 0.892 | 0.890 |
| 0.600 | 1.258 | 1.256 | 0.948 | 0.946 |
| 0.500 | 1.259 | 1.258 | 0.985 | 0.984 |
| 0.400 | 1.237 | 1.237 | 1.007 | 1.007 |
| 0.300 | 1.198 | 1.198 | 1.018 | 1.019 |
| 0.200 | 1.143 | 1.144 | 1.020 | 1.021 |
| 0.100 | 1.076 | 1.078 | 1.014 | 1.015 |
| 0.000 | 1.000 | 1.002 | 1.000 | 1.002 |
| -0.100 | 0.915 | 0.917 | 0.980 | 0.982 |
| -0.200 | 0.823 | 0.825 | 0.952 | 0.954 |
| -0.300 | 0.726 | 0.727 | 0.918 | 0.921 |
| -0.400 | 0.622 | 0.624 | 0.877 | 0.880 |

Table 2: The table compares our determination of $Z_\mu^{A,B}$ obtained by fitting equation 62 with those form [19].

# 4 Symmetries

## 4.1 Symmetries between $\mu$ and $-\mu$, $\lambda$ and $-\lambda$

The entanglement entropy is expected to possess several interesting symmetries in the scaling limit. The first such symmetry can be seen from the point of view of the perturbed homogeneous chain, where we have seen in section 3.3 that in the field theory Hamiltonian (48), translation of the cut by one site amounts to $\mu_R \rightarrow -\mu_R$. Of course this is true only to first order in $\mu_R$, but since the results in the scaling limit are valid in the limit $\mu_R \rightarrow 0$, $\ell \rightarrow \infty$ with $\mu \ell^{1-g^{-1}}$ finite, it is only this order that matters. Hence we conclude that, in the scaling limit:

$$\delta S(\mu) = -\delta S(-\mu) \tag{63}$$

The second symmetry is

$$\delta S(\lambda) = \delta S(-\lambda) \tag{64}$$

This follows from the discussion of the perturbation expansion around the split fixed point, and the fact that to all orders $\delta S$ was found to be an even function of $\lambda$. The relationships (63) and (64) are illustrated in Fig. 13a and Fig. 13b respectively. Note that, as emphasized above, the relationships are only expected to hold in the scaling limit, $\mu \rightarrow 0$ (resp. $\lambda \rightarrow 0$) and $L \rightarrow \infty$ with the appropriate combinations $\Theta_B$ (resp. $T_B$) finite. As commented earlier, the spread of the curves in the IR is due to the difficulty of reaching the scaling limit while being technically limited to relatively small values of $L$.

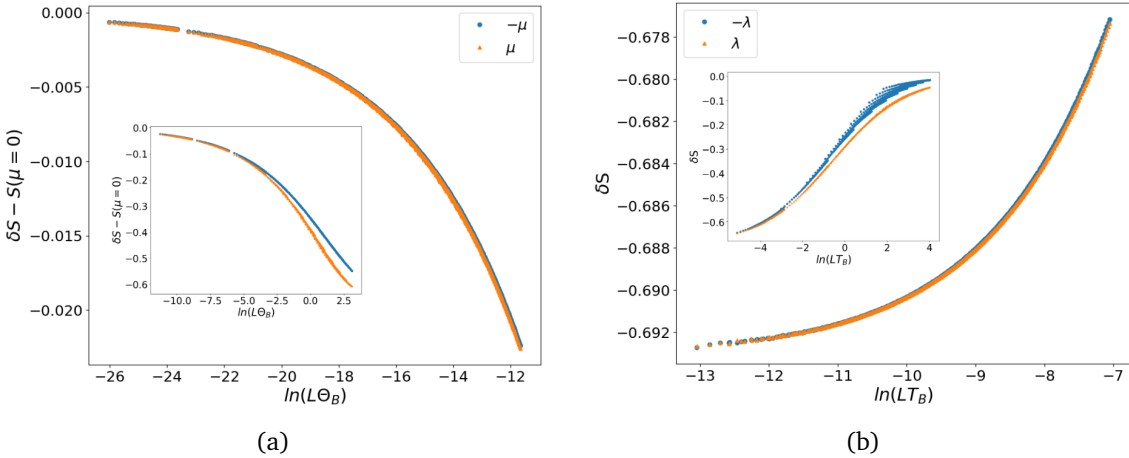

(a)                                                                        (b)

Figure 13: (a)Here $J_z$=0.5 and we compare $\delta S(\mu)$ and -$\delta S(-\mu)$ with Hamiltonian(42) when $\mu$ is very small and we are well in the scaling limit. (b) Here $J_z$=-0.5, and we compare $\delta S(\lambda)$ and $\delta S(-\lambda)$ with Hamiltonian(1) when $\lambda$ is very small. Insets are for a larger range of $L\Theta_B(LT_B)$.

## 4.2  Symmetries between $\lambda$ and $\frac{1}{\lambda}$

To see the third symmetry, imagine we consider a chain with $\lambda >> 1$ i.e. with a coupling between sites $\ell$ and $\ell + 1$ greatly enhanced. To facilitate the discussion we introduce a slightly more general Hamiltonian

$$
\begin{aligned}
H_\ell = {}& \sigma^x_{\ell-1}\sigma^x_\ell + \sigma^y_{\ell-1}\sigma^y_\ell + J_z\sigma^z_{\ell-1}\sigma^z_\ell + \lambda\left(\sigma^x_\ell\sigma^x_{\ell+1} + \sigma^y_\ell\sigma^y_{\ell+1} + \Delta\sigma^z_\ell\sigma^z_{\ell+1}\right) \\
& + \sigma^x_{\ell+1}\sigma^x_{\ell+2} + \sigma^y_{\ell+1}\sigma^y_{\ell+2} + J_z\sigma^z_{\ell+1}\sigma^z_{\ell+2}
\end{aligned}
\tag{65}
$$

where we have allowed for the coupling with amplitude $\lambda$ to have a different anisotropy $\Delta$. In the limit $\lambda >> 1$, the spins $\vec{\sigma}_\ell$ and $\vec{\sigma}_{\ell+1}$ are almost paired into a singlet. The Hamiltonian can then be replaced in this limit , by its first-order perturbation theory approximation

$$
H_\ell \mapsto -E_s + \sum_{t_i} \frac{|\langle s|H_\ell|t_i\rangle|^2}{E_s - E_{t_i}}
\tag{66}
$$

where the energies of the term coupling spins $\ell$ and $\ell + 1$ are $E_s, E_{t_i}$ respectively. For the singlet we have $E_s = -\lambda\left(\frac{1}{2} + \frac{\Delta}{4}\right)$ while the "triplet" now splits into states (for spins $\ell, \ell+1$) $|++\rangle$ and $|--\rangle$ with energies $E_{t_1} = E_{t_3} = \frac{\lambda\Delta}{4}$ and $\frac{|+-\rangle - |-+\rangle}{\sqrt{2}}$ with $E_{t_2} = \lambda\left(\frac{1}{2} - \frac{\Delta}{4}\right)$.

A straightforward calculation then gives, up to an irrelevant additional constant

$$
H_\ell \mapsto \frac{1}{\lambda}\left(\frac{\sigma^+_{\ell-1}\sigma^-_{\ell+2} + \sigma^-_{\ell-1}\sigma^+_{\ell+2}}{1+\Delta} + \Delta^2\sigma^z_{\ell-1}\sigma^z_{\ell+2}\right)
\tag{67}
$$

Observe that, while initially the modified bond was between sites $\ell, \ell + 1$, after this renormalization it is now between sites $\ell - 1$ and $\ell + 2$ which, after a relabelling starting as usual from the left, becomes between sites $\ell - 1$ and $\ell$. Hence we have exchanged the odd and even impurity problems. Notice also that the anisotropy of the Hamiltonian is not preserved in general. This only occurs in the XXX case when $\Delta = 1$, for which we recover an XXX Hamiltonian, and the coupling has gone from $\lambda$ to $\frac{1}{2\lambda}$ and in the XX case when $\Delta = 0$ for which we recover an XX Hamiltonian but the coupling has gone from $\lambda$ to $\frac{1}{\lambda}$.

The duality is best seen for Hamiltonian $H_B$ (7) which corresponds to $\Delta = 0$. In this case we expect, in the scaling limit

$$
\delta S(\lambda) = -\delta S\left(\frac{1}{\lambda}\right)
\tag{68}
$$

In general, since we have argued and checked that dependency of the $\delta S$ curve on the exact form of the modified Hamiltonian can entirely be absorbed into a redefinition of $T_B$, we expect the results for the problem and its dual to be identical (up to the exchange of odd and even) in the scaling limit. Moreover, in the case of Hamiltonians $H^A$ and $H^B$, the redefinition of $T_B$ can be obtained simply by the substitution $\lambda \to \frac{1}{\lambda(1+\Delta)}$. This relationship is illustrated in 14a, while the equation 68 is illustrated in 14b.

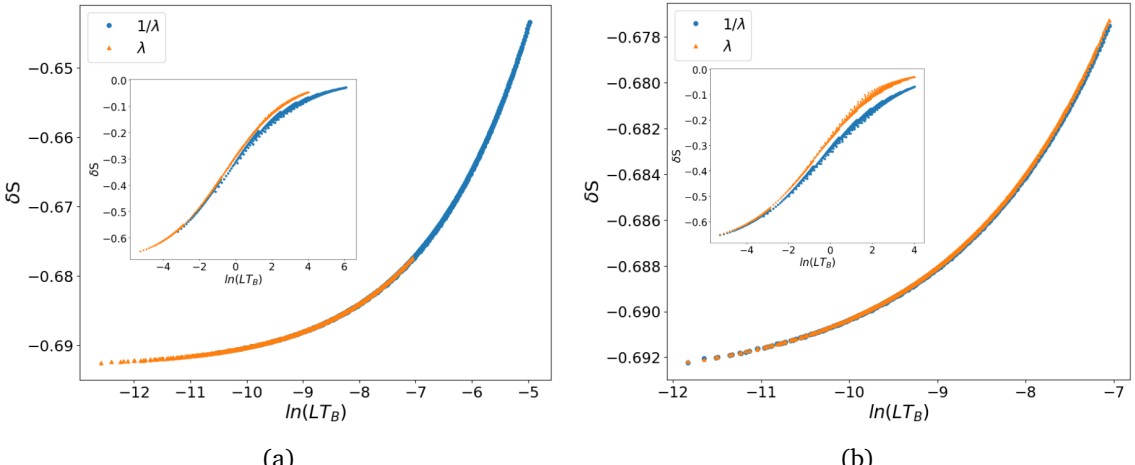

(a)

(b)

Figure 14: For these figures $J_z$=-0.5, and we compare $\delta S(\lambda)$ and $-\delta S\left(\frac{1}{\lambda}\right)$ when $\lambda$ is very small (and we are well in the scaling limit) (a) With Hamiltonian (1) and after the rescaling of the coupling by $\frac{1}{1+\Delta}$. (b) With Hamiltonian (7). In both cases the insert is for larger values of $LT_B$, where presumably the scaling limit is not fully reached.

# 5 Conclusions

While this problem originated in the context of physics near a boundary, the parity effects we unveiled occur as well in the bulk. Consider indeed a periodic system of length $L$ and a sub-interval of length $\ell$ connected on both sides to the rest of the chain by modified bonds as in (1,7) - this is illustrated in Fig. 15 below. The physics (flow towards a healed or split chain) is expected to be the same as near a boundary. We find that the entanglement for subsystems of even or odd length (the figure corresponds to the latter case) also differs by terms of $O(1)$. The details of these terms are a bit intricate, and we plan to discuss them elsewhere. For now we contend ourselves with the following observation. In the non-interacting case $J_z = 0$ and for two slightly different couplings $\lambda$ and $r\lambda$ with $0 < r < 1$, the difference $\delta S$ at large $L$ coincides, even in this periodic geometry, with the curve for the open geometry with a single modified bond $\lambda$: in other words, the weakest of the two modified bonds effectively behaves as if it were "opening" the system. While it is easy to understand this qualitatively (the system prefers to form valence bonds over the strongest bond), proving it analytically might be more difficult.

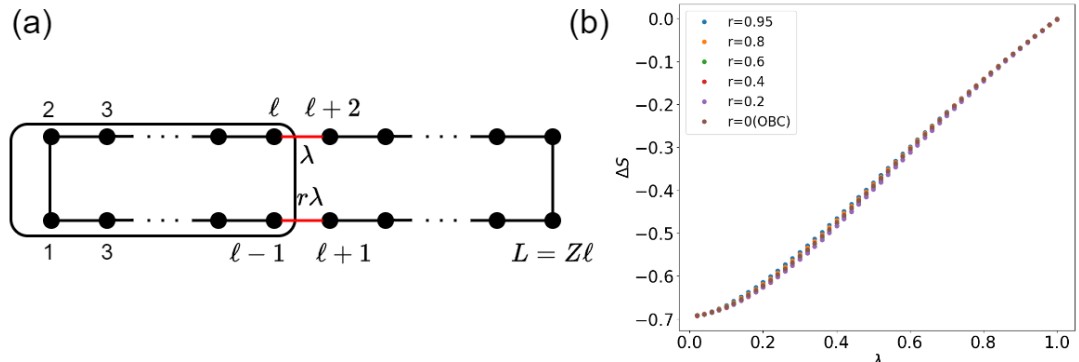

Figure 15: (a) Sketch of the PBC system. (b). Comparison of $\delta S$ for the systems with different ratio $r$, $r$=0 is corresponding to a OBC system, $J_z$=0.

# Acknowledgements

We thank H. Schloemer for related collaborations. HS thanks P. Calabrese and L. Capizzi for discussions. HS work was supported by the French Agence Nationale de la Recherche (ANR) under grant ANR-21- CE40-0003 (project CONFICA).

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
