# Peer review of "Entanglement parity effects in the Kane-Fisher problem"

_SciPost Physics_

## Round 1 · Referee Report · Anonymous (Referee 1) · 2024-7-25

Strengths

New effects and results on a timely problem

Weaknesses

Very minimalistic bibliography

Report

The manuscript "Entanglement parity effects in the Kane-Fisher problem" investigates the entanglement properties of a segment within an XXZ chain, with one entangling site connected to the rest of the system with a weak bond which is the well known Kane-Fisher problem. The other entangling site is left free. The authors identify a significant order 1 parity effects in the entropy, which they analyse through both numerical (DMRG) and analytical methods. A key finding is the resonance behaviour of the entropy difference between even and odd sites (which they call $\delta S$). such difference is a universal scaling function of $\ell T_B$, where the length scale $1/T_B$ is analogous to the Kondo length.

The paper addresses an important problem in the field of quantum many-body systems and entanglement theory. The interplay between entanglement entropy and impurities in XXZ chains is a topic of considerable interest in recent times. The introduction of a resonance curve in the entropy difference $\delta S$ and the application of conformal perturbation theory to analyse this quantity represent novel and fundamental contributions to the field. These results are likely to stimulate further research into similar impurity problems and the role of parity effects in quantum spin chains. The numerical results are robust and provide clear evidence of the predicted parity effects. Near the healed and split fixed points, the authors use conformal field theory and conformal perturbation theory to derive analytical expressions for $\delta S$. The perfect agreement between numerical and analytical results enhances the credibility of the findings.

The paper is well-written and organised. The introduction provides adequate background information. It could be published also in the present form. I have only three minor optional suggestions for improvement: 1. Some readers might benefit from a more detailed explanation of the terminology "healed" and "split" fixed points/chains in the introduction. As written now it is understandable to the specialists and it is a very easy concept that every reader of Scipost could understand. 2. The bibliography is extremely minimalistic with only the most relevant manuscripts cited. It could be integrated by adding some of the many references on entanglement for impurity problems both for interacting and non-interacting systems, including, but not limited to, some of the large literature on entanglement in Kondo problems. 3. Strangely enough the formulas appears all in bold in my PDF viewer. If this is a problem of the source file, it should be fixed.

Requested changes

See above

Recommendation

Ask for minor revision

---

## Round 1 · Referee Report · Anonymous (Referee 2) · 2024-8-1

Strengths

The analytical findings in this article are sound and in a reasonable agreement with the numerical data.

Weaknesses

Broader context of the problem is missing.
Results are not placed into context or perspective either.
Numerical details are not provided.
The manuscript is poorly structured with many errors and inconsistencies.
Figures a poorly explained.

Report

In the article ”Entanglement parity effects in the Kane-Fisher problem” the authors investigate the order one terms of the entanglement in a segment in an XXZ chain with a conformal impurity in the middle in the form of a weak bond. The authors use analytical methods based on conformal perturbation theory and check their predictions numerically with DMRG. This was done for two different scenario’s in which the chain is either healed or split in two under renormalization. A key finding is that the difference in order one terms of the entanglement entropy between even and odd sites, called $\delta S $, in XXZ chains, in accordance with the XX chain, shows resonance curves and universal scaling with the length of the segment $l$ and the Kondo temperature $T_B$. Their analytical findings are well performed, sound and agree well with the numerical data.
Although the key findings of the paper scientifically sound and could make a relevant contribution to the field of impurity problems in spin chains, akin Kondo problems, the manuscript is hardly accessible for non-experts. Also, focusing on the selected narrow problem that authors provide neither a broad-scope motivation behind the problem nor a wider perspective of their results. Presented results seems to me incremental compare to the previous works by authors.
To summarize, I cannot recommend the publication of this manuscript until the level of presentation is significantly improved (detailed comments are listed below). I also believe that the manuscript lacks originality (and undoubtedly it lacks a broad perspective) to justify its publication in SciPost Physics, though it might be suitable for SciPost PhysicsCore.

Requested changes

Below are my detailed comments: 1. Broader context of the results is missing. In large detail the results are derived and compared with numerical data. But there is no in-depth discussion, for instance, when numerical data demonstrate some deviation – what is the source of it, how to improve, etc? There is also no discussion on the broader impact of the results. 2. Introduction simply does not serve its role. No motivation is given, no context has been provided. When the authors write in the introduction “Think for instance of the Hamiltonian (1)” defined in the main body of the text it is extremely confusing. I understand that the authors worked on similar problems a lot and for them it is intuitive indeed to think about this Hamiltonian, but it is not as intuitive for the reader who have to see this Hamiltonian for the first time. In short, the manuscript requires reformatting and a better structured and clearer introduction to set the problem and a much more elaborate conclusion to place the findings into context, what they contribute to the problems mentioned above and proposals for further research. 3. What is the impact on specifically the Kane-Fisher problem never mentioned in the paper except in the introduction. 4. Numerical details are missing, making all numerical results non-reproduceable. This includes: Type of DMRG algorithm, bond dimension, convergence criteria, truncation error, number of sweeps, system sizes etc. 5. Lots of formulas are not formally introduced. The quantity of interest in this paper is the difference in the entanglement entropy between even and odd sectors $\delta S$. Besides the abstract it is nowhere introduced. Furthermore, this also holds for others important one. Take for example the definition of the constant $g$ appearing in the Affleck-Ludwig boundary entropy, shown in equation (2). This term includes boundary effects and is thus rather important for this study. Another example is the definition of the entropy in equation (34), which is not the well known formula for the Von-Neumann entropy and definitely requires the reference. 6. Details of the system is missing. An important part of the XXZ chain is the Luttinger Liquid that is realized for $|J_z|<1$. For someone that is known with either quantum spin chains or impurity problems it is obvious that the Luttinger Liquid is studied due to its free bosonic degrees of freedom and its corresponding logarithmic contributions to the entanglement entropy. But this is not clear for others. Furthermore, it puzzled me for a while whether the authors consider finite or semi-infinite segment length $l$. This should be clearly stated. 7. Some parts of the findings are not elaborately touched on. Take for example equation 67. When $\Delta \rightarrow -1$ the system approaches the phase boundary of the Luttinger Liquid with an exponential diverging Luttinger Liquid exponent $K$. This is reflected in this equation but is never described anywhere in the text. But this also hold for some behaviour in the numerical data. In figure 5 for example, there is a quite a bit of a discrepancy between the two curves. This has to be properly explained. 8. Important concepts and terms (though standard in the field) are not introduced, examples of which are healed and split, UV and IR, etc. 9. In section 3.3 the authors start to refer to equations in one of the references. It would simplify readers experience a lot if the authors provide these equations essential to understand part of the derivations. 10. Caption are far to insufficient in describing the figures. The simply do not give any idea on what is presented. 11. Sometimes the authors mention that results are checked numerically but not shown in the paper. Why are these not shown in an appendix? 12. In the healed case the Hamiltonian $H^B$ contains an extra term $J_z \sigma^z_l \sigma^z_{l+1}$ in comparison to the split case. There is no explanation, why it is so. 13. Some results of Table 1 are not mentioned in the text. 14. Variables and constants appearing in formulas are quite often not described. And there a few minor comments: 15. Quite a few typo’s in the text. Dots at the end of a sentence are missing. 16. Inconsistencies and errors in the formulas such as capitals $P$ in equation 32 , $|(0)+\rangle$ instead of $|(0)-\rangle$ in equation 38, same equation dash instead of double minus sign, etc. 17. As mentioned by another referee, all equations appear in bold. 18. Missing DOI in quite a few references.

Recommendation

Ask for major revision

---

## Editorial Decision

resubmitted